# OpenReview forum: "Beyond BFI: The CSI for Enhanced Reliability and Validity in Evaluating LLM Personality Traits"
_ICLR.cc/2026/Conference — Submitted to ICLR 2026_

### Official Review · Reviewer_R6L3 · 2025-10-28

**Soundness:** 2
**Presentation:** 2
**Contribution:** 1
**Rating:** 2
**Confidence:** 5

**Summary:**

This paper attempts to address the limitations of using human psychometric assessments, such as the Big Five Inventory (BFI), to evaluate the personality traits of Large Language Models (LLMs). To solve this, the paper introduces the Core Sentiment Inventory (CSI), a new evaluation instrument designed specifically for LLMs. Inspired by the Implicit Association Test (IAT), the CSI measures an LLM's implicit stance by asking it to associate a large inventory of 5,000 stimulus words with one of two evaluative poles: "comedy" (positive) or "tragedy" (negative). These associations are aggregated into three scores: Optimism (O_score), Pessimism (P_score), and Neutrality (N_score). The authors claim that, through extensive experiments, the CSI demonstrates significantly higher reliability and stronger predictive validity compared to the BFI.

**Strengths:**

* Important Research Direction: The paper tackles an interesting and increasingly important area of research: the study of LLM behaviors and the challenge of characterizing their personality-like traits.
* Clear Motivation: The authors' goal of creating a more robust, reliable, and valid evaluation system is promising. The paper identifies some known limitations in directly applying human-centric psychological tests like the BFI to non-human-like AI systems, such as model reluctance and prompt sensitivity.

**Weaknesses:**

* Lack of Psychometric Underpinning: The paper's central argument rests on discarding the BFI, but it fails to provide a psychometrically sound alternative. Earlier works selected the BFI not arbitrarily, but because it is a widely adopted and validated framework in human studies, believed to characterize actual, stable behavioral differences. The authors propose replacing this with a new three-dimensional O/P/N (Optimism/Pessimism/Neutrality) framework without sufficient theoretical or psychometric justification. This new framework feels arbitrary and it is not demonstrated why these dimensions are a valid, stable, or comprehensive way to characterize an LLM's behavioral tendencies.
* Non-Neutral Stimuli: The methodology's validity hinges on using neutral stimuli to probe a model's "implicit" stance. However, this claim is not convincing. A review of the authors' own examples in Table 2 reveals many chosen words are not neutral. For instance, the Top 5000 English words include "heal" and "bullying." This use of emotionally-related stimuli fundamentally renders the test results unconvincing as a measure of implicit personality.
* Interpretation of LLM Behavior (Instruction-Following vs. Inherent Traits): The paper seems to confuse what it is measuring. LLMs are, at their core, instruction-following systems whose behaviors can be significantly altered by system prompts and training.
* Consistency is Personality?: The paper claims high consistency as a strength. However, this is more likely a measure of the model's ability to consistently follow the simple instructions of the IAT-like task, not a measure of a stable, "inherent" trait.
* "Reluctancy" as a Feature, Not a Bug: The paper frames "model reluctance" (e.g., refusing to answer BFI questions with "As an AI, I do not have... beliefs") as a failure of the BFI. An alternative and more likely interpretation is that this reluctance is the model's inherent behavior, reflecting its alignment and safety training. The fact that the BFI triggers these guardrails may indicate it is probing targeted questions that reveal the LLM's core programming. The CSI, by design, seems to bypass this, and its "near-zero reluctance" may simply mean it is measuring a more superficial, less meaningful behavior.

**Questions:**

1. How do the authors justify their new method from a psychometric standpoint? Why should these be considered a valid or superior framework to the Big Five for characterizing model behavior, rather than an arbitrary, ungrounded classification?
2. Given that the stimulus set contains emotionally related words like "bully" and "heal", how does this align with the principle of avoiding explicit emotional words? How can the test results be considered a valid measure of "personality" rather than just a measure of common-sense association or alignment?
3. How can the authors distinguish between measuring a stable, inherent "personality trait" and measuring a model's "ability to follow instructions" consistently? Could the high consistency of CSI simply be an artifact of a simpler, more direct task?
4. The paper frames model reluctance as a failure of BFI. Could it not be argued that this reluctance is the model's 'inherent' behavior (i.e., its alignment)? Does designing a test (CSI) that avoids reluctance simply mean the test is measuring a more superficial, less meaningful behavior?

---

> ### Author Response · Authors · 2025-11-13
> **Response to Reviewer R6L3**
>
> We sincerely thank you for your time and effort in reviewing our paper. We respect your concerns, but we feel there may be some misunderstandings. We would like to offer the following clarifications.
>
> **On Question 1**: *How do the authors justify their new method from a psychometric standpoint? Why should these be considered a valid or superior framework to the Big Five for characterizing model behavior, rather than an arbitrary, ungrounded classification?*
>
> We would like to argue that our CSI framework is not an "arbitrary, ungrounded classification." To evaluate a psychometric tool from a psychometric standpoint, we follow the standard in this field: evaluating both its reliability [1] and validity [2].
>
> By applying this standard to both CSI and BFI, we provide a fair comparison. Our claim that CSI is superior in certain aspects is supported by the experimental results in Table 5 (showing higher reliability and near-zero reluctance) and Figure 4 (showing strong predictive validity). We believe this empirical evidence, grounded in established psychometric evaluation principles, justifies our method.
>
> [1] Cronbach, Lee J. "Coefficient alpha and the internal structure of tests." psychometrika 16.3 (1951): 297-334.
>
> [2] Messick, Samuel. "Validity of psychological assessment: Validation of inferences from persons' responses and performances as scientific inquiry into score meaning." American psychologist 50.9 (1995): 741.
>
> **On Question 2**: *Given that the stimulus set contains emotionally related words like "bully" and "heal", how does this align with the principle of avoiding explicit emotional words? How can the test results be considered a valid measure of "personality" rather than just a measure of common-sense association or alignment?*
>
> We acknowledge that some words, like "bully" and "heal," are not perfectly neutral, even after our filtering. We want to clarify that the examples shown in our paper were randomly sampled, not cherry-picked.
>
> The term "neutral word" is relative. Our method was guided by the analysis in SENTIWORDNET [3], which shows that nouns (0.944 objective score) and verbs (0.940) are significantly more neutral than adjectives (0.743) and adverbs (0.698). This is why we chose to build our stimulus set from nouns and verbs.
>
> Furthermore, we intentionally did not perform a stricter manual filtering to avoid introducing new human biases. For example, a word like "criticize" can be seen as negative by some but positive (as in "constructive criticism") by others. We deliberately kept such words to test the model's implicit association, which is central to our test's design.
>
> [3] Esuli, Andrea, and Fabrizio Sebastiani. "Sentiwordnet: A publicly available lexical resource for opinion mining." LREC. Vol. 6. 2006.
>
> **On Question 3:** *How can the authors distinguish between measuring a stable, inherent "personality trait" and measuring a model's "ability to follow instructions" consistently? Could the high consistency of CSI simply be an artifact of a simpler, more direct task?*
>
> We recognize that the reason for CSI's high consistency is worth discussing. Our primary goal in proposing CSI was to obtain more meaningful results.
>
> While high consistency might be partly due to the simpler task. The crucial point is whether these results are meaningful. We argue they are, because they translate directly to downstream task performance (as shown in Sec 4.3), reflecting the model's behavioral tendencies in real-world scenarios.
>
>
>
> **On Question 4:** *The paper frames model reluctance as a failure of BFI. Could it not be argued that this reluctance is the model's 'inherent' behavior (i.e., its alignment)? Does designing a test (CSI) that avoids reluctance simply mean the test is measuring a more superficial, less meaningful behavior?*
>
> We believe that if a model consistently gives meaningless answers (didn't provide a valid option) when using a test, common sense suggests the test is not effective.
>
> As for which test's results are more "meaningful," this is the exact question we answer in Section 4.3: VALIDITY ASSESSMENT. Our results show that CSI scores are highly correlated with model behavior in a downstream task.
>
> In contrast, as you can see in related works [4, 5], BFI scores for LLMs are almost never cross-validated with downstream tasks to prove they predict actual behavior. Given this, don't these facts support that CSI's results are more meaningful?
>
>
> [4] Jiang, Guangyuan, et al. "Evaluating and inducing personality in pre-trained language models." Advances in Neural Information
> Processing Systems 36 (2023): 10622-10643.
>
> [5] Huang, Jen-tse, et al. "On the humanity of conversational ai: Evaluating the psychological portrayal of llms." The Twelfth International Conference on Learning Representations. 2023.
>
>
> Thank you again for your effort and valuable feedback. We look forward to further discussion.

---

### Official Review · Reviewer_XoYK · 2025-10-31

**Soundness:** 1
**Presentation:** 2
**Contribution:** 1
**Rating:** 2
**Confidence:** 4

**Summary:**

The paper proposes a novel personality benchmark Core Sentiment Inventory (CSI) designed specifically for LLMs. The authors find that CSI captures behavioral nuances and variations between LLMs. They claim that CSI is more reliable than extant evaluation tools and that the high correlation between scores and outputs indicates strong validity for predicting LLM behavior.

**Strengths:**

This study identifies important short-comings with the llm personality assessment literature and argues for more reliable and valid evaluation tools.

**Weaknesses:**

The experimental design does not support the claim of reliability and validity of the CSI measure. For reliability, the authors test only two model types and they do not vary the prompts. If they conducted multiple trials, they do not report the repeated measures scores. For validity, they use the CSI words are seeds for generating stories, that are then evaluated for presence of the words used as seeds. This is plainly circular and does not prove that CSI is a valid measure of sentiment.
The main issue with using word associations as a proxy for personality assessment as some psychometric evaluations for human populations, is the unchecked assumption that the LLMs have stable internal representations. This assumption underlies the belief that words have unconscious associations and can be indicative of stable personality traits. Unfortunately, both of these claims remain to be proven. Thus, the soundness of word associations tasks for llm evaluations is questionable.
A quick inspection of the terms associated with comedy, tragedy, and neutrality show that they vary from model to model (Table 18, P25). With many of the same terms occurring in across the categories. This strongly undermines the construct validity. It is questionable from a face validity standpoint as well, what is the valence of "I" ???

**Questions:**

P3L142 "First, it alleviates test fatigue, a common issue in human-centric scales (e.g., 44 items in BFI, 100 in EPQ-R; see Table 1). With 5,000 items per language, CSI supports a broader and more inductive evaluation. Second, drawing on the implementation of the Implicit Association Test (Bai et al., 2025) on LLMs, CSI uses implicit assessment rather than direct self-report, which reduces refusal rates. Finally, CSI demonstrates stronger predictive validity, showing higher correlations between its scores and the sentiment of model-generated outputs"

=> How are these statements supported by your experimental design? Prima facie, I don't understand how a 5,000 item questionnaire would reduce test fatigue compared to a 44 item questionnaire. And if we're talking about LLMs, what does test fatigue have to do with anything? How do 5,000 items entail a broader and more inductive evaluation? The traits of optimism and pessimism are fewer and less behaviorally salient than neuroticism, openness, conscientiousness, extroversion, and agreeableness. That is to say, there is abundant empirical evidence linking these traits to other behavioral measures. Whereas the Big-5 have been shown to be stable traits, optimism and pessimism are affects that is they are transient and situational.

P7L364 "Second, we observe notable discrepancies in sentiment stance across languages. For example, GPT-4o shows minimal differences between English and Chinese, whereas LLaMA-3.1-70B displays a substantial divergence, with pessimism being dominant in Chinese (P_score of 0.47) compared to English (P_score of 0.30). This suggests that model behavior varies significantly across language
scenarios, a phenomenon that deserves deeper exploration."

=> Were tests of significance conducted? How do we know the means are significantly different?

---

> ### Author Response · Authors · 2025-11-17
> **Response to Reviewer XoYK (1/2)**
>
> We thank the reviewer for their critical assessment and for identifying the importance of developing reliable evaluation tools for LLMs. We really appreciate the opportunity to clarify our experimental design, validity framework, and theoretical grounding.
>
>
>
>
>
> **Reliability and Experimental Design**
>
> **Reviewer Comment:** *“The authors test only two model types... do not vary the prompts... do not report the repeated measures scores.”*
>
> **Response:** We respectfully clarify that our experimental setup is far more diverse than the reviewer perceived (detailed setup could be seen in **Experimental Setup** from line 312 to line 323):
>
> - **Diversity of Models:** We evaluated **five** distinct models (not two), including closed-source models (GPT-4o, GPT-4-1106, GPT-4-0125, GPT-3.5 Turbo) and open-source models (Qwen2-72B, LLaMA-3.1-70B).
>
> - **Repeated Measures & Consistency:** As detailed in the **Experimental Setup (Section 4)**, every model underwent repeated testing where we explicitly "shuffled the stimulus order in each iteration to assess the reliability level".
>
> - **Reporting Results:** We reported these repeated measures via the **Consistency Rate** metric in **Table 5**. The results show that CSI is significantly more stable than BFI. For instance, on the English dataset, CSI achieves an average consistency of **~79.9%**, whereas BFI averages only **~64.4%** .
>
> - **Prompt Variation:** While we use a fixed template for standardization, the stimuli are randomly sampled. Furthermore, we conducted extensive robustness checks involving **Cross-Lingual Prompting** (Table 14 ) and **Attribute Word Variation** (Table 15 ), finding that CSI maintains high reliability across these variations.
>
>
>
>
>
> **Validity**
>
> **Reviewer Comment:** *“They use the CSI words as seeds for generating stories, that are then evaluated for presence of the words... This is plainly circular.”*
>
> **Response:** We respectfully clarify that our validity assessment is **not** circular. Following the principle that validity refers to the extent to which a test measures what it is intended to measure[1] our analysis focuses on **whether CSI scores predict downstream behaviors**.
>
> In our approach, we prompt the model to write a story using CSI words. We then evaluate the sentiment expressed in that story—not merely the presence of the word itself.
>
> If CSI accurately captures a model's internal traits (e.g., a pessimistic view of "trains"), that traits should manifest as a tragic tone in a story about trains. The strong correlation (>0.85) observed in Figure 4 confirms that CSI scores are effective in predicting the sentiment of the model's generated content.
>
>
>
> [1]Messick, Samuel. "Validity of psychological assessment: Validation of inferences from persons' responses and performances as scientific inquiry into score meaning." American psychologist 50.9 (1995): 741.
>
>
>
>
>
> **Stability and "Test Fatigue"**
>
> **Reviewer Comment:** *“Unchecked assumption that LLMs have stable internal representations... I don't understand how a 5,000 item questionnaire would reduce test fatigue.”*
>
> **Response:** We apologize for any ambiguity and offer the following clarifications:
>
> - **Stability:** While LLM generation is probabilistic, "stability" in this context refers to the reproducibility of the evaluation metric. Our data (Table 5) empirically proves that CSI yields far more stable and consistent results than traditional BFI.
>
> - **Construct Validity & Variation:** The reviewer noted that terms like "I" have different meanings across models. We argue that this supports, rather than undermines, construct validity. First, it should be emphasized that these variations arise from differences between the models. This demonstrates CSI's discriminative power; it effectively captures the unique personality profiles of different models. For instance, one model might associate "I" with optimism, while another might associate it with neutrality. Certain atomic results might be less meaningful; however, these individual item-level emotional connections aggregate to form the overall personality of the model, reflecting the significance of each individual.
>
> - **Test Fatigue:** We apologize for the confusion. Our point regarding "test fatigue" was comparative: Human-centric scales (like BFI) are limited to ~44 items to prevent human fatigue. Because LLMs do not suffer from fatigue, we can expand CSI to **5,000 items**. This allows for a "broader and more inductive evaluation" by covering a vast spectrum of real-world concepts, rather than relying on a small set of psychological questions. We will clarify this phrasing in the revision.

---

> ### Author Response · Authors · 2025-11-17
> **Response to Reviewer XoYK (2/2)**
>
> **Optimism/Pessimism as Traits**
>
> **Reviewer Comment:** *“Optimism and pessimism are affects that is they are transient and situational.”*
>
> **Response:** We draw upon established psychological literature [2, 3]  which defines **Dispositional Optimism** as a stable personality trait reflecting generalized outcome expectancies, distinct from transient mood or emotion. In the context of LLMs, this maps to the model's "prior" probability of generating positive vs. negative completions in ambiguous contexts. In our experiments, this is a more stable characteristic, which might result from the distribution of its training data.
>
> [2] Scheier, Michael F., and Charles S. Carver. "Optimism, coping, and health: assessment and implications of generalized outcome expectancies." *Health psychology* 4.3 (1985): 219.
>
> [3] Scheier, Michael F., Charles S. Carver, and Michael W. Bridges. "Distinguishing optimism from neuroticism (and trait anxiety, self-mastery, and self-esteem): a reevaluation of the Life Orientation Test." *Journal of personality and social psychology* 67.6 (1994): 1063.
>
>
>
> **Statistical Significance**
>
> **Reviewer Comment:** *“Were tests of significance conducted? How do we know the means are significantly different?”*
>
> **Response:**
>
> We thank the reviewer for raising this point. We performed a paired t-test on the item-level scores (N=5000 items) to evaluate the cross-lingual discrepancies mentioned in Section 4.1. The results confirm that for Llama-3.1-70B, the Pessimism score in Chinese (Mean=0.47) is **significantly higher** than in English (Mean=0.30), with **p < 0.001**. This statistically confirms that language context significantly shifts the model's behavioral tendencies.

---

### Official Review · Reviewer_EDnX · 2025-11-01

**Soundness:** 2
**Presentation:** 3
**Contribution:** 1
**Rating:** 2
**Confidence:** 4

**Summary:**

The paper claims that BFI scores for LLMs are not stable and provides an alternative. The author critisise BFI methodology for "based on human-centered psychological theories" yet propose a new methodologe that is also "inspired" but a human centered work. At the same time the authors constantly mention "personality traits" as if LLMs do have personality.

**Strengths:**

The paper is rather well written, is in a relevant field and does address an important issue.

**Weaknesses:**

The subject of the reseach is not motivated convincingly. For example, human BFI scores also fluctuate (and this is well documented), and the level of fluctuations show in the opening figure is on par with human fluctuations. The methodology that authors propose, in my opinion, does not have any other nature that would make it more suitable for the computer systems per se. It is also not clear to which the proposed scoring mechanism would be beneficial for behaviour prediction that authors mention several times in the introduction.

**Questions:**

Here are some further works that could be informative for the scope of the work:

Sorokovikova A, Rezagholi S, Fedorova N, Yamshchikov IP. LLMs Simulate Big5 Personality Traits: Further Evidence. InProceedings of the 1st Workshop on Personalization of Generative AI Systems (PERSONALIZE 2024) 2024 Mar (pp. 83-87).

Pan X, Gao D, Xie Y, Chen Y, Wei Z, Li Y, Ding B, Wen JR, Zhou J. Very large-scale multi-agent simulation in agentscope. arXiv preprint arXiv:2407.17789. 2024 Jul 25.

Dong W, Zhao Y, Sun Z, Liu Y, Peng Z, Zheng J, Zhang Z, Zhang Z, Wu J, Wang R, Xu S. Humanizing llms: A survey of psychological measurements with tools, datasets, and human-agent applications. arXiv preprint arXiv:2505.00049. 2025 Apr 30.

Tshimula JM, Nkashama DJ, Muabila JT, Galekwa RM, Kanda H, Dialufuma MV, Didier MM, Kalonji K, Mundele S, Lenye PK, Basele TW. Psychological Profiling in Cybersecurity: A Look at LLMs and Psycholinguistic Features. InInternational Conference on Web Information Systems Engineering 2024 Dec 2 (pp. 378-393). Singapore: Springer Nature Singapore.

---

> ### Author Response · Authors · 2025-11-17
> **Response to Reviewer EDnX**
>
> We thank the reviewer for their time and valuable feedback. We appreciate the acknowledgment of the paper’s clarity and the importance of the research field. Below, we address the concerns regarding motivation, methodology suitability, and predictive validity.
>
>
>
> **Reviewer Comment:** *“Human BFI scores also fluctuate... and the level of fluctuations shown in the opening figure is on par with human fluctuations.”*
>
>
>
> **Response:** We respectfully acknowledge the reviewer's perspective regarding human psychometrics. We argue that CSI is more suitable than BFI for evaluating personality traits in LLMs.
>
> Our motivation is driven by the need for a more reliable metric for model development. As shown in **Table 5 (Section 4.2)**, CSI significantly outperforms BFI in reliability metrics:
>
> - **Consistency Rate:** CSI achieves up to a **45% increase** in consistency compared to BFI.
> - **Reluctancy Rate:** CSI reduces the non-response/refusal rate to **near zero** (e.g., for GPT-4, BFI refusal is 14.77%, whereas CSI is only 4.00%).
>
> In addition to consistent scores, BFI often suffers from model reluctance, which triggers safety guardrails (e.g., "As an AI, I do not have opinions"). This renders BFI structurally unsuitable for many models, regardless of score variance.
>
> We believe that even if BFI variance mirrors humans, a specialized tool (CSI) that eliminates refusal bias and improves consistency provides a more robust signal for AI developers.
>
>
>
>
>
> **Reviewer Comment:** *“The methodology... does not have any other nature that would make it more suitable for the computer systems per se.”*
>
> **Response:** We clarify that CSI is methodologically distinct from BFI in a way that specifically targets the **computational nature** of LLMs:
>
> - **Alignment with LLM Natures:** BFI relies on **explicit self-reports** ( eg, "Do you feel blue?"), which contradicts the RLHF safety training of many models (which are trained to deny having feelings). In contrast, CSI relies on **Implicit Association** (similar to token co-occurrence probability). This provides more explainable results, as LLMs may process these behaviors from their training corpus.
> - **Inductive "Bottom-Up" Design:** Unlike BFI, which imposes a "top-down" human-centric theoretical framework, CSI adopts a **"bottom-up" inductive design**. We first elicit **item-level implicit stances** towards specific stimuli and then aggregate these micro-behaviors into a comprehensive personality profile. This granular, data-driven approach provides a more detailed and mechanically faithful representation of the model's behavioral patterns than broad psychological questionnaires.
> - **Evidence of Suitability:** The suitability is best evidenced by the empirical results. The fact that CSI yields more consistency scores where BFI yields a higher refusal rate (Figure 1a vs. Table 3) demonstrates that CSI is better adapted to the current generation of LLMs.
>
>
>
> **Reviewer Comment:** *“It is also not clear to which the proposed scoring mechanism would be beneficial for behaviour prediction.”*
>
> **Response:** We apologize if this was not highlighted clearly enough in the introduction. We explicitly validate CSI's predictive capability in **Section 4.3 (Validity Assessment)** and **Figure 4**.
>
> - **Experimental Evidence:** We conducted a downstream "Story Generation" task. We found a strong positive correlation (**>0.85**) between a model’s CSI score (specifically the Pessimism/Optimism dimension) and the sentiment of the text the model generates in open-ended tasks.
> - **Practical Utility:** This demonstrates that CSI is not just an abstract score but a predictive indicator of how an LLM will behave in real-world applications (e.g., a model with a high "Pessimism" CSI score is statistically more likely to generate negative content).
>
>
>
> **Regarding Missing Citations:** We thank the reviewer for the suggested references. We will incorporate these valuable works into our "Related Work" section to better contextualize our contribution within the recent advancements in agent simulation and psychological profiling of LLMs.

---

### Meta-Review · Area_Chair_TjvP · 2026-01-06

**Summary:**

This paper proposes the Core Sentiment Inventory as an alternative to the Big Five Inventory for assessing personality in LLMs. By using an IAT-like paradigm, CSI measures optimism/pessimism/neutrality via word associations. The authors claim CSI is more reliable and predictive than BFI.

**Reviewer Concerns:**

(1) The core motivational argument remains weak. The rebuttal does not adequately justify why LLM “personality” assessment is meaningful if scores are not stable and why human-like fluctuations are problematic.

(2) Critical validity concerns persist. The story-generation validity check remains circular in design (using CSI words to generate text evaluated for sentiment).

(3) Fundamental issues are unresolved. CSI lacks a robust psychometric or theoretical foundation; the O/P/N framework appears arbitrary and is not shown to be comprehensive or stable for LLMs.

**Reviewer Scores:**

I don't think the reviewers will make signficiant change over the rating scores.

---

### Decision · Program_Chairs · 2026-01-26

Reject